# Coach–Athlete Relationships and Mental Health: An Exploratory Study on Former Female NCAA Student-Athletes

**DOI:** 10.3390/ijerph22111652

**Published:** 2025-10-30

**Authors:** Ashley R. Kernan, Michael R. Cope, Jonathan A. Jarvis, Mikaela J. Dufur

**Affiliations:** Department of Sociology, Brigham Young University, Provo, UT 84602, USA; ashleyrkernan@gmail.com (A.R.K.); jonathan_jarvis@byu.edu (J.A.J.); mikaela_dufur@byu.edu (M.J.D.)

**Keywords:** NCAA, collegiate athletics, mental health, public health, student-athlete well-being, female student-athletes, coach–athlete relationships

## Abstract

Female participation in NCAA athletics has grown significantly since the passage of Title IX—the 1972 U.S. federal law that prohibits sex-based discrimination in educational programs and activities receiving federal funding—yet much of the existing research continues to focus on male athletes, leaving important gaps in our understanding of women’s experiences in collegiate sports. One underexamined area with important public health implications is the role of coach–athlete relationships in shaping female athletes’ mental health, access to resources, and overall collegiate experience. This exploratory study draws on in-depth interviews with 19 former female NCAA athletes to examine how their relationships with coaches influenced their athletic careers, mental health, and perceptions of support. Participants represented a range of sports and competitive levels, allowing for variation in experiences across contexts. Findings reveal that coach–athlete relationships are not only central to performance and motivation but also serve as key sources of emotional, social, and material support—or, in some cases, stress and disengagement. The quality and impact of these relationships were shaped by competitive pressures, team dynamics, and institutional expectations. This study underscores the importance of relational context in understanding the broader landscape of female NCAA athletes’ experiences and suggests that coach–athlete dynamics merit greater attention in both research and athletic program development. These findings underscore the relevance of coach–athlete dynamics as a public health concern, particularly in relation to mental health and emotional well-being in competitive sports environments. Supporting healthier relational cultures in collegiate athletics is essential for promoting positive health outcomes among female student-athletes.

## 1. Introduction

During the 2023–2024 school year, approximately 540,000 student athletes across U.S colleges and universities participated in sports affiliated with the National Collegiate Athletic Association (NCAA) [1]. Student-athletes are an at-risk population in terms of how participating in competitive college sports affects their emotional, developmental, and academic experiences during an especially sensitive transitional period of the life course. While student-athletes experience multiple and intersecting stressors—academic, social, and personal—prior research also identifies the coach–athlete relationship as a particularly influential context in which these broader pressures are negotiated or intensified [2,3]. Our study examines this relational dynamic as one of several sources of stress that can shape female athletes’ overall well-being. These individuals face significant stressors in balancing academic and athletic demands, such as maintaining academic progress while experiencing intense training sessions, maintaining fitness levels, and dealing with injuries [4,5,6].

While men’s sports accounted for over 56% of participants, nearly a quarter of a million women (233,662) engaged in college athletics [1]. However, the literature outlining how sports participation might negatively affect athletes disproportionately emphasizes male athletes, marginalizing the experiences of their female counterparts. In this way, scholarship on college sports mirrors structural and institutional patterns that have privileged the experiences of male athletes over female athletes. This persistent gap in research obscures critical gender-specific patterns, potentially leaving female student-athletes particularly vulnerable to adverse events linked to sports participation [7,8]. As student-athletes constitute a growing and often vulnerable population within higher education, understanding the relational sources of psychological risk and protection is essential for developing effective public health responses.

As women’s and girls’ sport participation continues to grow, an especially significant gap remaining in the literature regards the role of the coach–athlete relationship in managing the challenges inherent in highly competitive sports. This relationship serves as more than a channel for skill development; it is a source of resources, emotional support, pressure, and access to playing opportunities. As a result, a power imbalance exists in relationships between college coaches and athletes. It is possible this power imbalance is exacerbated for female athletes, who are often coached by men who may not relate to their experiences [9,10] or by women who embrace masculine models of sport as a way to climb coaching ladders [11,12,13], while male athletes are almost exclusively coached by other men who have had similar athletic experiences. Existing literature highlights greater risks of adverse outcomes for female student-athletes, both compared to their female counterparts who are not athletes and to their male student-athlete peers; if coaches are either a key part of avoiding negative outcomes or a major source of causing those negative outcomes, understanding the coach–athlete relationship becomes especially important during the college years [7,8,14]. Examining this dynamic provides critical insights into the experiences of female athletes in higher education.

This study seeks to address this gap by investigating how coach–athlete relationships shape female student-athletes’ quality of life. In this study, we use quality of life (QoL) as a multidimensional concept that captures both the subjective and objective aspects of individual well-being within a specific social context. Following Long-Meek, Asay, and Cope [15], we treat QoL as an umbrella framework that integrates affective and evaluative dimensions of experience—linking emotional connection (often reflected in attachment) with cognitive appraisal (such as satisfaction and perceived support). Within this framework, QoL reflects how individuals experience and evaluate their overall sense of health, belonging, and fulfillment in relation to the social environments in which they participate. For female student-athletes, quality of life thus represents not only performance and physical health but also the relational, emotional, and institutional conditions that shape their collegiate experience.

The central research question guiding this exploratory study is: *How do coach–athlete relationships shape female student-athletes’ quality of life?* Specifically, we explore how former NCAA athletes perceive support, communication, and pressure within these relationships, and how these dynamics affect their well-being during and after their collegiate careers. To investigate these questions, we draw on nineteen in-depth interviews with former female student-athletes conducted between August 2023 and January 2024. Collegiate athletics function not only as training grounds for elite competition but also as institutional settings that shape long-term health trajectories. Within this context, the coach–athlete relationship plays a pivotal role in either protecting or undermining mental health. Power imbalances, chronic stress, and emotional abuse within these relationships may contribute to psychological distress, diminished coping capacity, and disengagement from help-seeking behaviors—factors widely recognized in public health literature as determinants of poor mental health outcomes. By examining coach–athlete dynamics as a relational determinant of health, this study contributes to the growing body of research focused on the social and structural contributors to mental well-being in high-performance settings.

### Coach–Athlete Relationships: Helpful, or Harmful?

A weighty stressor for female student-athletes (FS-A) within collegiate teams stems from coach power dynamics [3,16,17]. As a result, student athletes begin looking for quality relationships with coaches even before they are recruited to a specific team by considering the historical reputation of coaches in these programs [18]. The recruitment process itself often marks the beginning of power-laden coach–athlete dynamics. During this stage, prospective athletes are already navigating unspoken expectations around ideal athleticism, body type, and coachability—traits that can disproportionately affect female athletes as they attempt to align with program norms and secure roster spots [17,19]. The promise of scholarships and playing time often serves as implicit leverage in these early interactions, limiting athletes’ ability to advocate for themselves or question team culture before committing to a program [19]. These initial dynamics not only shape the foundation of the coach–athlete relationship but also contribute to a culture in which silence or compliance is perceived as necessary for long-term athletic opportunity. Furthermore, early access to competitive sports—often shaped by socioeconomic status—offers certain athletes a structural advantage in developing the kinds of cultural capital that make them more appealing recruits, reinforcing inequalities that carry into the college athletics experience [20].

Once student-athletes enroll in school and start both classes and competition, the coach–athlete relationship is crucial, shaping athletes’ collegiate experiences [11,16,21,22]. Coaches wield significant power, and they influence various aspects of student-athlete’s lives, including even their routine activities such as eating and sleeping [23,24,25]. Coaches also have authority over playing time, scholarships, and other key aspects of athletes’ sports opportunities, leading student-athletes to approach these relationships cautiously, fearing potential repercussions within their program [3,26]. Student-athletes face heightened stressors such as performance expectations, the fear of exclusion from competitions or meets, and the anxiety of letting down their coaches, staff, and teammates if they fail to meet fitness levels. This can all be impacted by the nature of the coach–athlete relationship.

While existing literature emphasizes the role of coaches in shaping student-athletes’ experiences, less research explores the broader impact coaches can have beyond practice settings, particularly in addressing the everyday hardships student-athletes may encounter [22]. Coaches using their power in an authoritarian approach can exacerbate stress, while those fostering supportive environments can mitigate stress and enhance well-being [22,27,28]. Athletes report greater satisfaction and feel their basic needs are met when they perceive strong coach–athlete relationships, fostering a sense of value and capability [22,29]. Strong coach–athlete relationships can be based upon closeness, understanding, commitment, and trust, which are associated with a positive sense of psychological safety for student-athletes in college [27,30]. Having a quality coach–athlete relationship can even increase student-athletes’ sports performance and reduce the likelihood of burnout for athletes throughout a competition season [30,31,32].

Conversely, negative experiences, including maladaptive coaching styles like emotional abuse, leave athletes feeling unwanted, stressed, and inadequate [22]. Alexander et al. [26] and Marsollier and Hauw [25] identified a troubling trend where emotional abuse like this is normalized under the guise of “tough love” to boost performance. As a result, some athletes report increased anxiety due to the fear of bullying and abuse by coaches [23,27,31]. Marsollier and Hauw [25] found athletes described examples of abuse in terms of negative and degrading comments about things like FS-A weight; in other studies, FS-A reported coaches pushed athletes beyond their mental and physical limits because they prioritized winning above athlete well-being [23,27]. Other abusive coaching patterns included such things as using the silent treatment, ignoring their athletes, and isolating them from others [3]. Such patterns describe coach–athlete relationships highlighted by destructive power dynamics where coaches wanted control of female athletes’ careers to boost their own reputations as coaches [23,25,27].

There is some evidence that female student athletes were less comfortable discussing mental health concerns with coaches compared to male student-athletes, potentially exacerbating feelings of powerlessness, which can be exacerbated in situations where coaches are abusive [25]. In cases of emotional abuse, athletes often see retirement or transferring as their only escape [3,25,26]. Furthermore, FS-A experiencing emotional abuse often require substantial processing to heal even after leaving harmful coach–athlete relationships [26].

## 2. Methods

### 2.1. Setting

This research involved interviews with former female NCAA student-athletes who competed and currently reside in the Continental United States. The focus on NCAA athletes stemmed from the NCAA’s significant role in overseeing collegiate sports, its standardized rules for athletes and coaches, the high-stakes nature of its competitions, and the availability of a large respondent pool [33].

Former athletes were intentionally selected rather than current NCAA competitors for both ethical and practical reasons. Active student-athletes are subject to institutional and media restrictions that limit their ability to participate in external research without prior university approval or media-relations oversight [34]. These restrictions could compromise confidentiality and inhibit open discussion of potentially sensitive experiences with coaches and athletic programs. Interviewing former athletes allowed participants to speak more freely about their collegiate experiences, providing richer, more reflective accounts of the coach–athlete relationship while minimizing any perceived risk of retribution or institutional consequences [35]. This approach aligns with established qualitative practices for studying sensitive topics within hierarchical or high-stakes environments.

### 2.2. Recruitment

Recruitment relied on existing relationships and snowball sampling, leveraging the first author’s position as a former student-athlete to facilitate entry into this hard-to-reach population. This insider status provided an *emic* understanding of participant experiences and helped identify initial contacts who, in turn, referred other eligible former athletes. This approach is consistent with other work and facilitated participant referrals within the target population [36,37,38]. Although the first author’s background provided important initial access and *emic* understanding of participant experiences, recruitment extended well beyond the first author’s personal network, ensuring a broader and more varied sample while maintaining participant confidentiality through peer referral. This approach aligns with established qualitative strategies for studying specialized or bounded populations while protecting participant confidentiality.

Eligible participants were former female NCAA athletes who had completed at least one year of collegiate competition in the USA. There were no restrictions on NCAA division or sport. Before participation, all respondents received an information sheet and signed informed consent forms outlining the study purpose, voluntary participation, and confidentiality protections.

### 2.3. Data Collection

Participants completed a brief pre-screening survey through Qualtrics to confirm eligibility. Nineteen one-on-one, semi-structured interviews were conducted via Zoom between August 2023 and January 2024, each lasting approximately one hour. Interviews were conducted in private settings to ensure comfort and confidentiality.

The interview guide was developed from themes identified in prior research on coach–athlete relationships, athlete well-being, and gendered experiences in collegiate sport [7,8,14,16,22,27]. A summary of the primary guide and corresponding thematic areas is provided in Appendix A (Appendix A). Questions addressed topics such as recruitment experiences, communication with coaches, perceptions of support and pressure, injury and recovery, mental health, and reflections on quality of life during and after college athletics. The guide was refined collaboratively among the research team to ensure clarity, relevance, and alignment with the study aims.

With participant permission, all interviews were recorded for transcription accuracy. Transcripts were auto-generated using Otter.ai and Zoom, then reviewed, corrected, and anonymized by the research team. During de-identification, all references to specific programs, universities, coaches, teammates, and facilities were removed to maintain confidentiality.

### 2.4. Data Analysis

Data analysis followed an exploratory, inductive approach aimed at deriving themes directly from the qualitative data. Members of the research team independently reviewed and coded the transcripts, after which we met to compare interpretations, discuss emerging patterns, and collaboratively refine themes through an iterative, data-driven process. This collaborative approach helped mitigate potential analytic bias by ensuring that no single researcher’s perspective— including the first author’s—dominated interpretation. Although we did not conduct formal intercoder reliability testing, we used consensus-based discussions [39] to ensure consistency and credibility in theme development. Transcripts were read multiple times and analyzed using the qualitative software Dedoose, (Version 9.0.107) with open codes developed, refined, and reorganized as analysis progressed until thematic saturation was reached [40,41]. Findings offer a detailed understanding of how coach–athlete relationships influence female student-athletes experiences.

## 3. Findings

Initial analyses revealed a novel finding: context matters. The intensity of athletic programs significantly shaped coach–athlete relationships and female student-athletes’ experiences. Since 1973, the NCAA has operated three divisions, with Division I being the most competitive and offering the largest budgets and resources [33,42,43]. For instance, in 2023, Division I Ohio State University reported nearly $275 million in operating expenses, compared to $11.3 million at Division II Ashland University, just 78 miles away [44,45]. While Divisions I and II provide athletic scholarships, Division III does not. Many aspiring athletes target Division I scholarships due to the opportunities they offer, but they also appear to find more fraught relationships with coaches in these more competitive settings. We report findings for two groups: student-athletes in more intense programs (MIP), emphasizing rankings and competition, and those in less intense NCAA programs (LIP) (See Table 1). Intensity levels were determined by NCAA division and participants’ descriptions of their experiences.

Given these findings concerning competitive level, respondents in our data who transferred across programs at different levels of competitiveness provided especially useful observations of patterns in both settings. Notably, all four FS-A who were transfers had transferred from more competitive programs to less competitive programs.

The average age of the 19 female student-athletes interviewed was 24, with most holding bachelor’s degrees. Our sample included 89% Division I athletes and 10% Division III athletes. Gymnasts comprised 53% of participants, followed by swimmers/divers at 32%, and the remaining 15% included a softball player, a lacrosse player, and a volleyball player. Half of our participants spent 4+ years on their teams, while 21% transferred during their NCAA careers (Table 1). While this means that half of our sample comprised FS-A from one sport, patterns of findings did not differ across sports, increasing our confidence in their reliability.

We report three important patterns of findings below: relationships during recruitment, coach–athlete relationships during FS-A’s time on campus, and the lasting impact of these relationships after competition.

### 3.1. The Coach–Athlete Relationship Before Coming to Campus

#### Dreams Become Reality: Recruitment Promises and Lived Experiences on Campus

Many respondents dreamed of participating in collegiate athletics from a young age. Watching collegiate sports growing up, they envisioned exhilarating experiences. However, upon arriving on campus, many found their experiences differed from the promises made by coaches during recruitment.

FS-A recruited to LIPs reported positive recruiting processes. Reagan, a gymnast, shared that her recruiting experience aligned with her eventual team experience, noting the coaches were approachable and honest. All LIP athletes interviewed shared similar sentiments, describing coaches as forthright during recruitment and finding their campus experiences matched their expectations. This honesty fostered strong, positive coach–athlete relationships that persisted throughout their collegiate careers.

In contrast, FS-A recruited to MIPs often experienced unmet expectations, citing discrepancies between recruitment promises and the reality of the program. Darcy, a gymnast, noted, “*Official visits or unofficial visits go, where everyone has to say great things and seem nice and make the school look all shiny and squeaky clean and then you show up and it’s just the exact opposite*.” Violet, a gymnast added:


*The coach is trying to sell themselves just like we’re going to be your second parents, we’re so nice and we are a big family here and obviously, a lot of times that ends up not really being the case.*


All student-athletes at MIPs reported experiencing some form of what Jenna described as being “catfished.” Ten FS-A explicitly noted that coaches were welcoming and supportive during recruitment, describing the team as family-like, but all ten reported a stark reversal upon arriving on campus. Violet, a gymnast, shared her experience at an MIP, stating: “There was so much manipulation and mental abuse… it was definitely not the experience I thought I was going to be getting.” Sam, another gymnast, and Brenna, a swimmer/diver, described a common sentiment of recruitment promises not being met:


*Even on my recruiting experience, they promised a certain gymnastics atmosphere, a team atmosphere, even a coaching atmosphere, and I feel like none of those promises came through. I just feel like it was just very toxic.*



*On the recruiting trip, they’re showing all these amazing outlets that they have to help student-athletes do well in classes, but then, you try and utilize them, and if it interferes with the sport in any way, you’re met with a little bit of hostility.*


Many FS-A felt that coaches painted recruitment as the beginning of the “best years of their lives,” but their experiences often fell short. This false narrative created trust barriers with coaches throughout their collegiate careers. Audrey, a softball player, reflected: “You realize that you shouldn’t be going through what you got”. The discrepancy between recruitment promises and reality fostered a stressful, often toxic environment for MIP student-athletes, leaving many disillusioned with their collegiate experiences. This contribution creates understanding that even if student-athletes begin searching for potential college coaches before they are recruited and look at the historical history of coaches’ reputation [18], there is more of a story to be told from the promises to reality that many of these FS-A experienced.

### 3.2. The Coach–Athlete Relationship on Campus

#### 3.2.1. ”Machines for School” or Student-Athlete?

As student-athletes progress through their collegiate careers, the influence of their coaches becomes increasingly significant. This relationship is often intense, as FS-A navigate injuries, training, performance, and various coaching and motivational tactics. Alongside these challenges, FS-A must also manage typical college stressors, such as demanding coursework, unsupportive professors, and difficult roommates, while facing additional pressures from their coaches. These interactions, whether positive or negative, have long-lasting effects, leaving some FS-A disappointed or questioning if they would repeat the experience.

Athletes at LIPs generally reported positive and supportive relationships with their coaches. Reagan, a gymnast, reflected: *“You always have that one coach who you’re grateful that came into your life, and she was definitely that coach for me.”* Similarly, Susan, another gymnast, stated that her coach *“did their best to support us.”* These athletes described more positive experiences overall, often trusting their coaches with both athletic and non-athletic concerns. A key distinction was how LIP coaches prioritized athlete recovery. Injuries were handled with understanding and a focus on long-term health rather than immediate team performance. Violet, a gymnast who transferred from a MIP to a LIP, shared:


*If the trainers told them, ‘Oh this person is injured,’ they [coaches at LIP] wouldn’t even question it; they would be like, you’re doing basics today…. They genuinely didn’t want people injured, and it was cool.*


This approach contrasted sharply with her MIP experience. The emphasis on athlete well-being at LIPs reflected a less hierarchical coach–athlete relationship, prioritizing the student-athlete as a whole rather than solely as a competitor.

The LIP experience was characterized by bidirectional coach–athlete relationships, where coaches demonstrated care that extended beyond athletics to the personal lives of their athletes. Willow, a gymnast, noted that at her LIP, coaches would ask about personal matters, such as, *“What did you do over the weekend? Or, ‘Oh, you went home to see family; how’s mom and dad?’”* These interactions align with the literature on the positive influence of supportive coaches on student-athletes during college [22,27]. Interviewees described LIP coaches as understanding of injuries, supportive of athletes as individuals rather than solely as competitors, and genuinely invested in their well-being. Of the eight student-athletes interviewed from LIPs, four had transferred from MIPs, offering unique perspectives. All four spoke more favorably about their transfer schools. Violet, a gymnast, shared:


*It was not like a facade ever; they were dead serious about wanting it to be a positive experience… Everybody supported each other, and the coaches [at the LIP] were so supportive.*


In contrast, student-athletes at MIPs reported coach–athlete relationships that were less supportive, less personal, and more focused on team performance than individual well-being. These relationships were often described as complicated, conditional, and negative. Audrey, a softball player at a MIP, stated: *“(The coach) wasn’t supportive. He was really manipulative…he was not at all looking out for our well-being; if he did, it was very probably surface level or fake.”* Notably, these commonly described negative relationships with coaches which may have actually hindered rather than helped overall athletic performance [27,32,46].

All student-athletes at MIPs (15) described their coach–athlete relationships negatively. Many FS-A shared similar sentiments about their relationships with their coaches. Willow, a gymnast who eventually transferred to an LIP, stated, *“(Coaches at the MIP) didn’t really care about how you’re doing”.* Brenna, a swimmer/diver, echoed, “He (coach) just didn’t understand…; he (coach) just didn’t care.” Sam, a gymnast, explained, “Mental health and physical health—he didn’t really have a care in the world for (them)… just pretty negative,” while Holly, a swim/diver, expressed “I was getting discriminated against.” These unsupportive relationships were perceived as hierarchical, focusing solely on athletic performance rather than the athletes’ holistic well-being. Many FS-A noted that their coaches showed little interest in getting to know them beyond the sport, nor in taking an active role in healthy development. Sam recounted:


*I was told I was heavy, pathetic, and so my body image just shot down through the drain, too. So, I feel I was really struggling, and I would struggle at practice even. I talked to a coach about it at one point, and he told me, you know what, you just need to mature and learn how to control your emotions, so he just had no care that I was struggling when I was reaching out.*


Sam’s experience reflects a broader negative pattern in coach–athlete relationships at MIPs, where coaches’ comments and behaviors create challenges for student-athletes. When FS-A sought help, they often felt blamed for their feelings, with coaches framing these challenges as deficits in personal development. This abdication of the teaching and guiding aspects of coaching aligns with previous findings by Haan and Norman [17] and Stirling and Kerr [3], who identified damaging and hierarchical patterns in coaching. The in-depth interviews presented here provide further evidence that FS-A perceived coaches as leveraging their power in hierarchical, unbalanced relationships, prioritizing outcomes and performance over holistic support. This contrasts with the broader framing of collegiate sports as opportunities for comprehensive growth [47]. While there is ongoing debate about the role of coaching, particularly the balance between sport-specific skills and broader mentoring responsibilities [48], our findings indicate that FS-A at MIPs in particular, felt they were not receiving training or development in educational or life skills from their coaches.

Some FS-A at MIPs felt that coaches paid lip service to the broader goals of collegiate sports, but as Jocelyn, a swimmer/diver, noted, *“their actions spoke louder than their words.”* In MIP settings, FS-A often felt viewed as “machines for the school,” as Willow, a gymnast, described, or as athlete-students rather than student-athletes. This perception aligns with Dufur and Feinberg’s [49] findings, where professional football players were treated as expendable cogs in the football industry. The FS-A we interviewed echoed similar sentiments, bridging their experiences with the literature on male athletes, who often report feeling disposable or part of an impersonal system designed for others’ profit [49,50,51].

#### 3.2.2. Injured? Suck It up and Keep at It

Navigating the coach–athlete relationship became particularly challenging when athletes dealt with injuries. Many FS-A reported that coaches were unsympathetic, often denying the severity of injuries, discouraging adequate recovery, and resorting to guilt, shame, or punishment. Such behaviors, including yelling or assigning additional consequences, left many athletes feeling uneasy about attending practice.

Given the physical demands of high-level competition, injuries were common among the FS-A in our sample. More than half recounted instances where coaches were dismissive or unsupportive of their injuries. Many faced additional repercussions, such as threats of reduced playing time, diminished importance on the team, or even the loss of scholarships. These threats, viewed by FS-A as credible, highlighted the hierarchical nature of the coach–athlete relationship, where coaches held control over critical resources [26]. The pressure to train or compete while injured was a recurring theme. Many athletes feared practicing in pain, knowing that coaches expected immediate participation despite their physical condition. Sam, and Violet, gymnasts at a MIP, both recalled an expectation that was prevalent among injured student-athletes:


*There was a girl who had a broken finger, and she still had practice bars every day because the coach didn’t really care…They expected us to be back at practice the next day even if you were hurting or you weren’t feeling well or you were injured.*



*I was in the ER, and I still had to come to the gym the next day, like five hours after I left the emergency room… and two days later, alright, [the coaches said] you need to get back on it.*


Some student-athletes described feeling guilted by coaches for being injured or seeking treatment. Brenna, a swimmer/diver, explained: “one or two days [when injured], you would be respected, and then, the coaches often would expect you to be at practice… Prioritizing your body was often met with a lot of backlash and just kind of guilt.” Such experiences illustrate the guilt-tripping and shame many FS-A encountered in more intense programs when injured. These feelings were exacerbated when coaches encouraged athletes to internalize blame for their injuries. Coaches frequently avoided using formerly injured athletes in competitions even after recovery, with one athlete even threatened with being cut from the team due to an injury that prevented training. FS-A interpreted these actions as punitive, further straining coach–athlete relationships. Unsurprisingly, relationships rarely recovered to their pre-injury quality when athletes felt punished for pain or circumstances beyond their control. These findings add a perspective from female student-athletes and aligns with current research suggesting coaches’ focus tends to be on program success rather than holistic athlete development [26].

#### 3.2.3. “Are You Good Enough?” Coaches’ Use of Mind Games as Manipulation

Many FS-A reported that coaches employed maladaptive tactics, such as playing mind games during practice, to create a sense of urgency and anxiety. Sam, a gymnast, shared: “If someone would fall in a competition, she (coach) would threaten to take them out the next week…. She would psych girls a lot just with how she would threaten their spot[s].” More than 50% of FS-A described experiencing some form of mind games, with varying levels of intensity. Among those at MIPs, all but one reported encountering this behavior.

Susan, the only gymnast at a LIP who experienced mind games, suggested that coaches may not have been intentionally “messing with their heads,” but acknowledged it still occurred. In contrast, three athletes at MIPs described their coaches as withholding coaching or giving them the “silent treatment” if dissatisfied with practice performance or if athletes were late due to scholastic responsibilities. Darcy, a gymnast, recounted her coach’s use of “manipulation and mind games” to threaten her during practice, including imposing unattainable standards not required of other teammates to compete. She explained:


*It felt like you’re giving me these expectations, but you’re making it that so I cannot meet them, you don’t want me to get to that point, and it was done in a way to convince you that you’re not good enough to do the skills you’ve been doing for 5–10 years of your career.*



*These mind games extended beyond practice and competition. Violet, a gymnast, described how her coach leveraged others to manipulate her:*



*It was just creating so much division between me and everybody else in my life, feeling like everybody’s watching me but in a nefarious way—not watching me because they care, watching me because they’re like, ‘Oh, she’s going to mess up,’ and it was really bad. It got to the point where I was super paranoid everywhere I went.*


Some coaches may contend that strategies described by student-athletes as detrimental to the coach–athlete relationship and athlete well-being are well-intentioned efforts to enhance performance. However, the student-athletes interviewed consistently reported that tactics such as mind games and perceived punishments not only failed to improve competitive performance but also caused emotional and physical harm. These actions often led to irreparable damage to the coach–athlete relationship. Notably, three FS-A transferred from their universities due to the negative treatment they experienced. This finding aligns with existing literature, which suggests that, in some cases, the only way to escape such abuse is to leave; for NCAA student-athletes, this often involves quitting the sport or transferring to another program [26].

#### 3.2.4. Tough Love: Coaches’ Counterproductive Use of Verbal and Physical Punishment

While some FS-A reported that coaches used mind games as a motivational tool, another theme that emerged that negatively affected coach–athlete relationships was yelling and the use of physical punishment at MIPs. Audrey, a softball player [MIP], described instances where coaches employed extra workouts as a form of punishment: *“He would make us run, I would pass out, he would just keep making everyone run… I would push myself until I couldn’t do it.” FS-A reported that verbal punishment was also commonly used at MIPs. Hazel, a gymnast, explained that coaching sometimes evolved “more into an aggressive form of coaching…depending on the coach, some forms of verbal abuse… they would yell all the time, and there were some comments made that… break people down.”* Both Audrey and Hazel expressed that yelling and punishment became “very normal for us,” as Hazel noted. Similarly, Jocelyn, a swimmer/diver, described the team environment as “an anxious environment” due to coaches’ frequent yelling, which created widespread tension among athletes. Violet further stated that “everyone was just miserable.” FS-A exposed to verbal or physical punishment rarely recounted positive experiences with their coaches, severing a critical source of support.

Research highlights a history of punitive coaching behaviors affecting both male and female athletes [22]. Former student-athletes from various sports have reported that such coaching styles often felt normalized, perpetuating maladaptive practices [26]. Although these punitive actions are common, athletes frequently described experiencing significant stress as a result of these negative encounters. Willow, a gymnast, recounted that her experiences at MIP escalated to the point where “the coaches were very verbally abusive.” Other interviewees shared similar accounts from their MIP, describing efforts to endure the mistreatment as it became increasingly normalized. However, they were left without adequate support to cope with these challenging experiences.

These instances of punishment suggest that coaches were perceived as exhibiting authoritative power. Whether intentional or not, this behavior reflects an exploitation of the inherent power imbalance in coach–athlete relationships, raising concerns about the extent to which these relationships can genuinely benefit FS-A. While coaches have the opportunity to foster an environment that promotes growth and support, the interviewees indicated that many prioritize their success and methods over the individual development of FS-A. This misuse of power creates an environment perceived as abusive, one that fails to provide FS-A with the support necessary for their well-being and growth.

### 3.3. Long-Term Effects of Coach–Athlete Relationships After College

#### 3.3.1. Lasting Impact on Student-Athletes

Most research concerning coach–athlete relationships, for both male and female athletes, focuses on the time during which the athlete is actively competing. Our respondents, however, described important connections between the quality of the relationships with their coaches and their long-term mental and physical health after their competitive days were over—and these effects were not uniformly negative. While discussing the aspects of these relationships that caused harm, some women were also able to reflect on valuable lessons and personal growth. Melanie, a swimmer/diver, emphasized that *“it’s going to be an overall positive experience with something that can contribute to your life in a good way if you choose it to be that way.”* Similarly, Susan, a gymnast, shared, *“I think going through all this kind of taught me also to stand up for myself a little bit more.”* Some FS-A in MIPs rationalized their negative treatment by focusing on positive experiences despite the challenges they faced. Audrey, a softball player, expressed that these hardships made her “more resilient,” while Zoey, a gymnast, stated that she chose to “take away more of the positives.” These FS-A reclaimed agency over their NCAA experiences, striving to ensure that the negative aspects did not overshadow the benefits of their participation. Many highlighted the lifelong lessons learned from their sport, even when their coaches contributed negatively. Courtney, a gymnast, emphasized, “It taught me how to trust in myself and really find myself.”

Despite some student-athletes’ ability to find positive takeaways from the sport, many were left with lasting negative impressions of their treatment on NCAA teams and their relationships with coaches. Violet, a gymnast, described how the manipulation and mental abuse she experienced at her MIP continue to affect her:


*The kind of anxiety and paranoia has kind of stayed with me… being out in public feeling I’m being watched all of the time; I do still feel that. I also feel I struggled in friendships and relationships for a while, too, because when you have your coach manipulating you and playing these mind games and then yelling at you and saying, it’s, oh, just because I care about you and I want you to succeed, it’s kind of hard to understand after that, after being treated like that for so long… and from an outsider’s perspective, it seems obvious but when you are like experiencing it for so long, it really does become hard to tell.*


Violet’s experience continues to impact her today. While she sought a bidirectional, supportive interpersonal relationship with her coach, she instead encountered manipulation and abuse. Similarly, Holly, a swimmer/diver, described how her cumulative NCAA experiences and her coach–athlete relationship influenced her life beyond sports:


*I would say I’m a lot less trusting. I’m a lot more anxious and warier of meeting new people and working with new people. I was like, the shoe just got to drop, the other shoe is going to drop, something’s going to happen. I’m going to, if I get close to this person, they’re going to leave. If I like you, you’re going to leave. It’s just going to happen.*


Holly also stated, *“I do see myself sometimes flashing back to [University] athletics; something will trigger it, and I’m like, dang, that really happened to me.”* These reflections underscore the prolonged impact coach–athlete relationships can have on athletes. Brenna, a swimmer/diver, shared, “I still get really frustrated” when reflecting on her NCAA experience, while Darcy, a gymnast, noted, *“The whole experience definitely still resonates today; I still feel it a lot… The details are kind of blurry, and trauma blocking it to a stance.”* Athletes explained how they are still coping with these emotions even after they have retired from their sports. This suggests that coach–athlete relationships can leave a lasting effect on athletes. Several FS-A reported that experiencing abusive or unsupportive coach–athlete relationships had a significant impact on their mental health, resulting in enduring challenges such as anxiety, depression, paranoia, sadness, stress, anger, and perfectionism [27,30,31]. Our findings extend previous research finding that coach–athlete relationships can impact the student-athlete’s overall health during competition to describe the ways these experiences shape the rest of their lives.

#### 3.3.2. “I Almost Have a Regret for Doing It and I Am Thankful I Did It”: Framing and Reframing the Choice to Participate in NCAA Athletics

Given the challenges associated with strained coach–athlete relationships and their lasting negative effects, we explored whether FS-A would choose to participate in NCAA sports if allowed to decide again. Responses from the 19 FS-A varied. Most FS-A who competed at LIP indicated they would choose to participate again, citing stronger coach–athlete relationships that made the decision an “easy answer.” In contrast, four FS-A from MIP stated they would make the same choice, not because they enjoyed their experiences, but because those experiences shaped who they are today. Robin, a gymnast, and Jenna a Volleyball player, expressed:


*I feel like it sucked, and not all of it sucked, like parts of it that sucked that I hated, but in the end, I like am grateful for, I guess. So, I’d probably say, yes.*



*People always ask if I know now what would have happened would I still choose to go to there and I always tell them yes, because even though it was super hard and didn’t really go with my plan, I just learned so much from it, and grew as a person from it. So, I was really grateful for the hard experiences.*


For other athletes, all of whom competed at MIPs, the decision was more difficult to assess. Some, like Sam, a gymnast, expressed regret about competing at the NCAA level and indicated they might not have made the same choice again:

Looking back, I almost have a regret for doing it and I am thankful I did it. I feel like I learned a lot. I grew a lot as a person. I just feel like it was more of a put down than it was a prize and a success, especially more for my mental and physical health.

Many athletes grappled with an internal struggle to frame their NCAA experiences, acknowledging both the difficulties they endured and the personal growth they achieved. Notably, this growth was often attributed not to coaching but to their resilience in overcoming the challenges their coaches imposed. In reframing their experiences with a more positive outlook, some FS-A exhibited signs of toxic positivity. Toxic positivity involves suppressing negative emotions and experiences in favor of reframing them as positive outcomes [52,53]. By interpreting their struggles as growth through adversity, these FS-A sought to downplay the painful aspects of their experiences and assign worth to what they endured. However, a final group of athletes, all of whom competed at MIPs, were unequivocal in stating they would not make the same decision to participate again. Darcy, a gymnast provides the first of four FS-A provided compelling examples of this:


*It’s not worth it… if I would have known that it was going to be like, you can still find a way to do the sport you love without having to do it to that capacity and put yourself through the living hell that was NCAA gymnastics.*


Similarly, Holly, a swimmer/diver stated that, *“Wow, I really had to put up with this… I refuse to associate myself with [University] athletics… I didn’t like my experience at athletics, kind of hate them, kind of hate NCAA.”* Brenna, a swimmer/diver, stated that, *“I don’t think I would ever go back and do the same situation again and things always look better on paper.”* Lastly, Jocelyn, a swimmer/diver, noted that, *“Honestly, I just wanna say no, cause it makes me feel really gross inside… my anxiety could not handle it… I don’t think I would do that again, knowing what was gonna come.”*

Interestingly, while some FS-A reframed their NCAA experiences as formative, shaping them into the individuals they are today; this reframing often proved insufficient to mitigate the negative effects on their mental health. Many FS-A stated they would not repeat their NCAA experiences due to the significant toll on their mental and physical well-being. Those in MIPs frequently reported being treated like cogs in a machine, lacking the supportive and encouraging interpersonal relationships with coaches that they had hoped for. Instead, they described environments characterized by hostility, abuse, and a lack of support. Robin, a gymnast, reflected, “I think your innocence is taken away and you wonder, too,… is it like this at other schools or was this just my experience at the school?” The enduring negative effects of these experiences were so profound that two FS-A indicated they would never allow their future children to participate in the same sports. Others expressed that their NCAA experiences served as lessons on the kind of influence they did not want to have on their children. Holly, a swimmer/diver, shared, *“Everything that they [coaches] did to me, I’m not doing to them. I’m a lot more aware of my experiences and how I’m going to not do that to the little kids.”*

## 4. Discussion

This exploratory study examined how coach–athlete relationships shape female student-athletes’ quality of life during and after their collegiate careers. The findings reveal that program context, particularly the competitive intensity of athletic programs, played a central role in shaping the quality and character of these relationships. Across experiences, athletes described coach–athlete interactions as key sources of either support or strain, influencing their motivation, well-being, and long-term perceptions of collegiate sport.

It is important to note that we do not suggest the coach–athlete relationship is the primary or sole cause of stress among student-athletes. Rather, consistent with prior literature, we identify it as a key relational stressor that operates alongside other academic, social, and personal factors. Participants described this relationship as a central channel through which broader pressures were experienced—sometimes mitigated through support and at other times exacerbated by conflict or power imbalance.

Although the majority of participants in this study competed in individual sports such as gymnastics and swimming/diving, the patterns observed appear to reflect relational dynamics rather than sport-specific factors. Participants across both individual and team contexts described similar experiences of pressure, support, and control linked to their interactions with coaches. This suggests that the challenges reported by athletes stem less from the nature of individual competition and more from coaching style, communication patterns, and the broader relational climate cultivated within athletic programs. The consistency of these themes across sports reinforces that the key determinant of athletes’ well-being was the quality of the coach–athlete relationship, not the type of sport itself.

Our findings from this exploratory study revealed that context plays a critical role. FS-A in LIP reported better quality and more supportive relationships with coaches, which positively affected their overall experiences. These findings add deeper context into the current literature on how positive coaching can enhance a student-athlete’s quality of life [2,21,27,30,31,54,55,56].

In contrast, FS-A in MIP described feeling that coaches prioritized their performance over their well-being, perceiving the relationship as purely instrumental. These athletes often lacked relationships in which they felt comfortable communicating their needs to supportive coaches. Instead, they experienced predominantly unidirectional relationships that left them feeling disheartened. Respondents consistently emphasized that coaches in MIP treated them as mere cogs in a machine, mirroring findings by Dufur and Feinberg [49] regarding college football players’ interactions with professional coaches at the NFL draft combine. Our findings demonstrate that coaches treating relationships with athletes in purely instrumental terms is not limited to professional sports; rather, such destructive patterns are common in college sports, in spite of expectations that coaches will be part of sports helping development in young people.

Coaches in MIP were perceived as focusing primarily on winning competitions and meets, whereas coaches in LIP were described as fostering bidirectional relationships that supported FS-A development and growth. Coaches in NCAA programs wield considerable power, influencing not only the athletic performance of student athletes but also their academic, physical, and mental well-being [23,27]. While the pressure to win is an inherent aspect of the coaching profession, coaches also have the opportunity to positively shape athletes’ experiences or leave lasting trauma. The lack of perceived concern from MIP coaches had a profound and enduring impact on the interviewees, many of whom stated that the trauma from their NCAA experiences continues to resonate with them. It is imperative to amplify and prioritize student-athletes’ voices to better understand the dynamics within NCAA teams that might otherwise appear outwardly ideal. This understanding is essential for addressing the challenges faced by FS-A and fostering environments that support their holistic development. These findings additionally allow for further insight on understanding the specific extent in which female student-athletes are affected both positively and negatively. As Senel et al. [30] have indicated, in a more positive environment, athletes are more likely to indicate psychological safety [27]. Whereas, in negative environments, athletes can be negatively impacted both mentally and physically [23,25,31]. Lastly, most literature focuses on the impacts of male athletes. While this research suggests that negative environments can impact them negatively, our research furthers the investigation by examining female college athletes in predominantly female sports and how the impact of these environments affect them so drastically and how those female student-athletes in MIPs are negatively affected by their coach–athlete relationships [8].

Our findings highlight the need to recognize coach–athlete relationships as not only performance-relevant but also as critical to athletes’ long-term health outcomes. Given the well-documented challenges in college mental health broadly—and the added pressures facing female student-athletes in high-stakes programs—these relationships may act as either protective or pathogenic influences [14,22,27,29,30]. From a public health perspective, improving relational dynamics in athletic settings is not merely about better team culture; it is a structural intervention with potential to reduce anxiety, trauma, and long-term psychosocial harm [7,8,25,27]. This aligns with emerging research emphasizing relational determinants of health and the need for institutional accountability in sport and educational environments [16,19,26].We encourage future research and prevention efforts that integrate public health frameworks into athlete mental health promotion, with specific attention to gendered dynamics and power hierarchies in coaching environments.

While our findings contribute to the literature on FS-A experiences and the significant influence of coach–athlete relationships, several limitations must be acknowledged. First, the study’s findings are not necessarily generalizable to all NCAA sports. The sample consisted of 19 former FS-A, including only four athletes who had competed in LIPs. Second, due to positionality and the use of snowball sampling, the study disproportionately included FS-A who were gymnasts or swimmers/divers. Interestingly, even though all of our respondents played “non-revenue” sports, which some literature has suggested might be associated with less pressure, those at MIPs still reported that expectations around success and winning drove the nature of their relationships with coaches. Our approach here could be extended to more visible and better compensated sports, such as basketball, to see how additional visibility affects coach–athlete relationships. Finally, the retrospective nature of the interviews represents both a notable limitation and a notable strength. Participants were former student-athletes reflecting on their collegiate experiences, which may be subject to recall bias or influenced by the passage of time. At the same time, this retrospective approach revealed previously undescribed patterns of long-term effects on athlete development.

To address these limitations, future research should aim to include a larger and more diverse sample of FS-A from a broader range of NCAA sports, including those NCAA sports that are revenue-generating sports. Expanding the sample size and variety of sports could provide a more comprehensive understanding of coach–athlete relationships and student-athlete experiences. Additionally, a longitudinal panel study design, following student-athletes from recruitment through and beyond their collegiate careers, could offer deeper insights into their evolving experiences. Targeted research on student-athletes who have transferred during their collegiate careers could shed light on the unique challenges and dynamics they face. Finally, as American college athletics have changed in the last few years to allow athletes to earn money from their Name, Image, and Likeness (NIL), this may alter the way athletes are treated by coaches or their expectations to perform. Examining how financial opportunities affect athletes at MIPs is of interest, as the pressure for coaches and athletes may be intensifying.

While acknowledging its limitations, this exploratory study sheds light on the prevalent power dynamics within coach–athlete relationships. It further highlights the potential influence of NCAA program intensity on shaping these relationships. The findings demonstrate that coaches can significantly impact FS-A experiences, either positively or negatively, during their athletic careers. This study provided 19 FS-A with a platform to share their experiences and feel heard. While some interviewees, primarily from LIP, reported positive experiences, many student-athletes from MIP expressed feelings of disappointment and trauma. Some attempted to reframe their negative experiences using toxic positivity, yet they still felt the enduring effects of those challenges. Despite the pressure to achieve competitive success, coaches have a choice in the techniques they employ—either fostering a supportive and positive environment or creating impersonal and potentially traumatizing experiences. The narratives shared by these women illustrate that the treatment they received from their coaches heavily influenced whether their NCAA experience was empowering or detrimental, truly a “make or break it” for these women.

## 5. Conclusions

This study highlights the significant role coach–athlete relationships play in shaping the collegiate experiences of female student-athletes (FS-A), particularly in relation to the competitive intensity of their programs. Through interviews with 19 former NCAA athletes, we found that program context, especially the distinction between more intense (MIP) and less intense programs (LIP), shaped whether coaches were perceived as supportive mentors or controlling figures focused primarily on performance. Athletes in LIPs described more positive, relationally supportive coaching that extended beyond athletics, while those in MIPs often reported hierarchical, transactional dynamics marked by manipulation, emotional harm, and pressure to compete through injury. Importantly, the effects of these relationships extended beyond college athletics, influencing FS-A’s long-term mental health, personal relationships, and perceptions of trust and self-worth. While some respondents reframed their experiences as opportunities for growth, others expressed enduring distress or regret. These findings underscore the critical need to attend to the relational culture of NCAA programs, particularly in highly competitive contexts, and to hold coaches accountable not only for athletic outcomes but for the holistic well-being of the student-athletes in their care.

## Figures and Tables

**Table 1 ijerph-22-01652-t001:** Descriptive Statistics for Former NCAA Female Student-Athletes (*N* = 19).

Variable	*n*	%
Demographics		
Level of Education		
Some college	3	15%
Bachelor’s degree	14	74%
Master’s degree and beyond	2	11%
Age		
18–21	4	21%
22–25	9	47%
26–30+	6	32%
NCAA Sport		
Gymnastics	10	53%
Swim/Dive	6	32%
Softball	1	5%
Lacrosse	1	5%
Volleyball	1	5%
NCAA Division		
Division I	17	90%
Division III	2	10%
Intensity of Program		
Less Intense	4	21%
More Intense	15	79%
Years on team		
1 year	1	5%
2 years	4	21%
3 years	4	21%
4+ years	10	53%
Transferred		
Yes	4	21%
No	15	79%
NCAA Division on transferred team		
Division I	4	100%
Intensity of Program		
Less Intense	4	100%
Years on transferred team		
Less than 1 year	1	25%
2 years	1	25%
4 years	2	50%

## Data Availability

Due to concerns surrounding confidentiality, redacted data presented in this study are only available upon reasonable request from the corresponding author.

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
