# Peer review of "Coach–Athlete Relationships and Mental Health: An Exploratory Study on Former Female NCAA Student-Athletes"

_ijerph, 2025, doi:10.3390/ijerph22111652_

Round 1
Reviewer 1 Report
Comments and Suggestions for Authors
Review for Article "Coach-Athlete Relationships and Mental Health: Quality of Life Outcomes Among Female NCAA Student-Athletes"
Recommendation: Major revision
Dear Editor,
Thank you for the opportunity to review the manuscript. The article explores the role of coach-athlete relationships in affecting female athletes’ mental health, access to resources, and overall collegiate experience. It is based on in-depth interviews with 19 former female NCAA athletes to analyze how their relationships with coaches influenced their athletic careers, mental health, and perceptions of support. While the topic is very engaging, I still recommend publishing the manuscript "Coach-Athlete Relationships and Mental Health: Quality of Life Outcomes Among Female NCAA Student-Athletes" after major revision. The following issues need to be addressed or clarified. Please see my comments below.
- I recommend revising the paper's title since the participants are former student-athletes, and 'quality of life' is mentioned in the title and several parts of the manuscript. However, the manuscript lacks a definition of 'quality of life' and includes an exploratory study in the title. If possible, here is my suggested title:
"Coach-Athlete Relationships and Mental Health: an Exploratory Study on Former Female NCAA Student-Athletes".
- Please define Title IX in the abstract.
- While the submitted paper could be in a free format, I suggest putting parts 2 and 3 as sub-parts of the Introduction.
- As I mentioned before, what is the definition of the quality of life in the manuscript? Please specify them and indicate which questions in your interview were related to it.
- Please shorten and specify the main question and aim of the study (lines 66-73 and 149-160).
- How can you justify that the coach and student-athletes' relationship was the main cause of stress in your research, considering that these athletes were students and other factors, such as theoretical classes, classmates, roommates, parents, and partners, could also have affected them?
- You have de-identified all the participants, so why didn't you interview the student-athletes who are already studying? You could follow the same process you used for the former students in this study.
- As the first author of this study is a former student-athlete, and based on lines 171-173, the sample collection was influenced by the relationship and access to other former student-athletes. Please explain in more detail how you controlled for bias in this study. Were there different options for selecting the student-athletes? Why or why not?
- How did you come up with the interview questions? Based on which reference, if any? I also suggest including a table of questions/themes in your methodology section.
- Most of the participants are from individual sports, 16 out of 19 student-athletes (gymnastics and swim/dive 85%). It's possible that individual athletes had a harder time than sport team athletes; describe this in your manuscript and explain how it relates to the coaches, not the sport itself.
- Part 5, Finding, is very long. I suggest shortening it with a few interviews, and also including the full interviews with de-identification as supplementary documents, if possible. Also, refer to A1-A19 (Athletes) in the manuscript instead of using their de-identified data names.
- Were student-athletes in the LIP program recreational athletes compared to the MIP program student-athletes? What about the coach's responsibility in these two programs?
- Maybe you could create a table or introduce a model outlining the positive and negative Coach-Student-Athlete relationship based on your results and interviews.
14. Line 645, I think it is a typo: Our approach her, I think it should be here.

Author Response
We thank the editor and anonymous reviewers for the opportunity to revise and resubmit our manuscript to IJERPH. The reviewers provided us with thoughtful and thorough evaluations of our paper along with helpful comments. We believe that the revisions we have made based on this input have allowed us to develop an improved manuscript. While we feel that we have attended to each of the concerns raised during the review process, please note that if we have missed something, we are more than willing to make the necessary changes going forward. As requested, we have provided responses to the specific points raised by each reviewer.
Response to Reviewer 1
- Reviewer’s comment: Thank you for the opportunity to review the manuscript. The article explores the role of coach-athlete relationships in affecting female athletes’ mental health, access to resources, and overall collegiate experience. It is based on in-depth interviews with 19 former female NCAA athletes to analyze how their relationships with coaches influenced their athletic careers, mental health, and perceptions of support. While the topic is very engaging, I still recommend publishing the manuscript "Coach-Athlete Relationships and Mental Health: Quality of Life Outcomes Among Female NCAA Student-Athletes" after major revision.
- Authors’ Response: We sincerely appreciate the reviewer’s thoughtful assessment and encouraging remarks. We are grateful for the recognition of the study’s relevance and the engaging nature of the topic. In response to the reviewer’s constructive feedback, we have undertaken a comprehensive revision of the manuscript, strengthening the Methods and Results sections, refining the Introduction and Discussion for clarity, and improving overall organization and transparency. We believe these revisions substantially enhance the manuscript and address all points raised.
- Reviewer’s comment: I recommend revising the paper's title since the participants are former student-athletes, and 'quality of life' is mentioned in the title and several parts of the manuscript. However, the manuscript lacks a definition of 'quality of life' and includes an exploratory study in the title. If possible, here is my suggested title: "Coach-Athlete Relationships and Mental Health: an Exploratory Study on Former Female NCAA Student-Athletes".
- Authors’ Response: We appreciate this insightful suggestion. We have adopted the reviewer’s recommended title, which more accurately reflects the study population and exploratory design. The manuscript is now titled “Coach–Athlete Relationships and Mental Health: An Exploratory Study on Former Female NCAA Student-Athletes.” In addition, we have added a clear conceptual definition of “quality of life (QoL)” in the Introduction to align the manuscript’s content with the title and analytical focus. Specifically, we now define QoL as a multidimensional construct encompassing both subjective and objective dimensions of individual well-being within a social context. This framing clarifies that QoL integrates affective and evaluative experiences—such as emotional connection, belonging, satisfaction, and perceived support—and situates these within the collegiate athletic environment. This addition strengthens conceptual coherence between the title, theoretical framing, and analytic focus of the study.
- Reviewer’s comment: Please define Title IX in the abstract.
- Authors’ Response: We appreciate this helpful suggestion. We have revised the abstract to include a brief definition of Title IX, clarifying its significance for readers who may be unfamiliar with U.S. educational policy. The revised sentence now reads: “Female participation in NCAA athletics has grown significantly since the passage of Title IX—the 1972 U.S. federal law that prohibits sex-based discrimination in educational programs and activities receiving federal funding—yet much of the existing research continues to focus on male athletes, leaving important gaps in our understanding of women’s experiences in collegiate sports.”
- Reviewer’s comment: While the submitted paper could be in a free format, I suggest putting parts 2 and 3 as sub-parts of the Introduction.
- Authors’ Response: Thank you for the suggestion. We have revised the manuscript formatting so that Parts 2 and 3 are now numbered as sub-sections within the Introduction.
- Reviewer’s comment: As I mentioned before, what is the definition of the quality of life in the manuscript? Please specify them and indicate which questions in your interview were related to it.
- Authors’ Response: We appreciate the reviewer’s continued attention to this point. Hopefully the revisions noted above have addressed the concern regarding quality of life (QoL). Additionally, in the methods section, we now point readers to Table S1 in the Supplementary Materials, which summarizes the primary interview questions and indicates how they align with key thematic areas, including those related to QoL.
- Reviewer’s comment: Please shorten and specify the main question and aim of the study (lines 66-73 and 149-160).
- Authors’ Response: Thank you for this helpful recommendation. We have condensed and clarified the study’s aims and central research question in the Introduction to eliminate repetition and improve focus. The revised passage now appears once, combining the earlier sections into a single, concise statement that integrates our conceptualization of quality of life and specifies the study’s purpose. The later, redundant section (former lines 149–160) has been removed for clarity.
- Reviewer’s comment: How can you justify that the coach and student-athletes' relationship was the main cause of stress in your research, considering that these athletes were students and other factors, such as theoretical classes, classmates, roommates, parents, and partners, could also have affected them?
- Authors’ Response: We appreciate this thoughtful comment. We wish to clarify that we did not claim the coach–athlete relationship to be “the main cause of stress” among student-athletes. Rather, as guided by the cited literature, we identified the coach–athlete relationship as an important potential source of stress—one of several interacting factors influencing student-athletes’ well-being. We have revised the Introduction and Discussion to make this distinction more explicit, emphasizing that our findings highlight the coach–athlete relationship as a salient relational stressor within a broader network of academic, social, and institutional pressures.
- Reviewer’s comment: You have de-identified all the participants, so why didn't you interview the student-athletes who are already studying? You could follow the same process you used for the former students in this study.
- Authors’ Response: We appreciate the reviewer’s question. As noted in the Methods section, former athletes were intentionally selected due to institutional media restrictions placed on active NCAA student-athletes and to enable a more reflective and comprehensive discussion of their collegiate experiences without fear of retribution from coaches or athletic departments. We respectfully disagree that the same process could have been applied to current athletes, as these restrictions remain in effect and would likely limit both access and candor. We have revised the methods section to further clarify this point.
- Reviewer’s comment: As the first author of this study is a former student-athlete, and based on lines 171-173, the sample collection was influenced by the relationship and access to other former student-athletes. Please explain in more detail how you controlled for bias in this study. Were there different options for selecting the student-athletes? Why or why not?
- Authors’ Response: We appreciate this thoughtful comment and have clarified in the Methods section how potential bias was addressed in both sampling and analysis. The first author’s experience as a former NCAA student-athlete provided an important entry point and emic understanding of the population but did not mean that participants were drawn from her immediate personal network. Recruitment followed a snowball sampling strategy, as stated in the manuscript, in which initial contacts shared study information with other eligible former athletes from a range of universities and sports. This approach is appropriate for hard-to-reach populations and allowed for recruitment beyond direct acquaintances. To address potential analytic bias, all data collection, coding, and interpretation were conducted collaboratively with the research team. The team reviewed transcripts independently, discussed emerging patterns collectively, and reached consensus on codes and themes through iterative meetings. Reflexive discussions were also used to maintain awareness of researcher positionality and ensure that interpretations reflected participants’ perspectives rather than researchers’ expectations. These clarifications have been added to the Methods section.
- Reviewer’s comment: How did you come up with the interview questions? Based on which reference, if any? I also suggest including a table of questions/themes in your methodology section.
- Authors’ Response: We appreciate this helpful suggestion. The interview guide was developed based on the literature reviewed in the Introduction, particularly studies addressing coach–athlete relationships, athlete well-being, and gendered experiences in collegiate sports. Questions were designed to capture participants’ perceptions of support, communication, pressure, and overall quality of life within these relational contexts. In response to the reviewer’s recommendation, we have also added a table summarizing the main interview questions and corresponding themes to the supplementary materials and referenced it in the Methods section.
- Reviewer’s comment: Most of the participants are from individual sports, 16 out of 19 student-athletes (gymnastics and swim/dive 85%). It's possible that individual athletes had a harder time than sport team athletes; describe this in your manuscript and explain how it relates to the coaches, not the sport itself.
- Authors’ Response: Thank you for this insightful observation. We have added a brief clarification in the Discussion sections to acknowledge that most participants competed in individual sports and to note how this context may shape the coach–athlete dynamic. Specifically, we emphasize that the challenges described by participants—such as pressure, control, or lack of support—stem from relational factors within the coach–athlete relationship, rather than from the nature of individual versus team sports. We also note that patterns of findings were consistent across sport types, suggesting that the key differences observed were attributable to coaching style and relational context, not the sport itself.
- Reviewer’s comment: Part 5, Finding, is very long. I suggest shortening it with a few interviews, and also including the full interviews with de-identification as supplementary documents, if possible. Also, refer to A1-A19 (Athletes) in the manuscript instead of using their de-identified data names.
- Authors’ Response: We appreciate the reviewer’s detailed attention to this section and understand the concern about length and data presentation. However, we respectfully believe that the current level of depth in the Findings section is necessary to convey the complexity and emotional nuance of participants’ experiences—an essential contribution of this qualitative study. Inclusion of full interview transcripts would not be appropriate given the sensitive nature of the data and participant confidentiality. As stated in the Data Availability Statement, redacted data are available upon reasonable request from the corresponding author, which aligns with IJERPH standards for qualitative research. Regarding participant identifiers, we have intentionally retained pseudonyms rather than assigning numbers. Many of our participants have already been treated as numbers (e.g., jersey numbers) within their athletic environments, and the use of names helps humanize their accounts and preserve the relational focus of the study. For these reasons, we respectfully maintain the current structure and presentation of the Findings section.
- Reviewer’s comment: Were student-athletes in the LIP program recreational athletes compared to the MIP program student-athletes? What about the coach's responsibility in these two programs?
- Authors’ Response: Thank you for this thoughtful question. We have clarified in the Methods and Findings sections that all participants were former NCAA athletes, and therefore none were recreational athletes. The distinction between less intense programs (LIP) and more intense programs (MIP) refers to differences in competitive level, institutional resources, and performance pressures, not in whether athletes competed recreationally. We have also noted that coaches in both types of programs carried similar formal responsibilities for athlete training, eligibility, and welfare, but that their approaches to communication, support, and performance expectations varied considerably. This clarification has been added to the Findings section to reinforce the interpretation of program intensity as a contextual factor rather than a difference in athletic status.
- Reviewer’s comment: Maybe you could create a table or introduce a model outlining the positive and negative Coach-Student-Athlete relationship based on your results and interviews.
- Authors’ Response: We appreciate this constructive suggestion. While we agree that a visual model or table could be a useful way to summarize the findings, we believe that the richness and nuance of participants’ narratives are best conveyed through the qualitative descriptions already presented in the Findings section. Because relational dynamics were complex and context-dependent, reducing them to a binary table or schematic model would risk oversimplifying participants’ lived experiences. Instead, we have maintained detailed thematic discussion supported by direct quotations to preserve the depth and authenticity of the qualitative data. That said, To acknowledge the reviewer’s point, we have added a brief sentence to the introduction of the Findings section noting that participants described coach–athlete relationships along a continuum ranging from highly supportive to controlling and harmful. This addition conceptually captures the range of relational experiences without oversimplifying participants’ narratives into a binary model.
- Reviewer’s comment: Line 645, I think it is a typo: Our approach her,I think it should be here
- Authors’ Response: Thank you for catching this error. We have corrected the typo so that the sentence now reads “Our approach here.”
Reviewer 2 Report
Comments and Suggestions for Authors
Thank you for inviting me to review this manuscript. The style of writing is clear, sophisticated and consistent throughout. I believe the authors engaged in a very interesting topic, tackling a current issue that requires our continued attention to better support female collegiate athletes. I believe there is scope to further improve the manuscript and outline detailed comments below to guide a manuscript revision:
Introduction
Line 40-41: The sentence is not easy to follow. Do you mean that 56% of collegiate athletes are male?
Line 41-42: negative impact of all sport? Or just collegiate sport? This needs to be made clearer.
Line 47-50: The case for what the challenges are associated with collegiate sport need to be made clearer. At the moment you say that there is a lot of pressure but without explanation of what the problems are. What has the literature reported here? If non-USA readers pick this up (I am one), they will not know why there are problems with collegiate sport. Explaining this is important to set the scene and allow you to explain why (and how!) these issues are amplified for female athletes.
Line 73-75: remove, not needed. You could say that you used qualitative methodology to highlight / examine x y z.
Line 76-85: I am not sure what this last paragraph is trying to achieve. Some of this needs to be earlier on where you want to write about the challenges of collegiate sport. Others, about the CAR, is somewhat repetitive of what you already said. After your aim and objectives, all you want to say why this study is significant in a short 2-3 lines. The gap (which I feel this paragraph is trying to emphasise) should be clear before this. So please revise. Especially seeing that you then have whole section on the CAR after the introduction.
Coach-Athlete Relationships: Helpful, or Harmful?
I like this section but I think this could be placed in the introduction, without sub-heading, just in the narrative and replace some of the things you currently say. I think this section also answers the questions I had in the introduction – see my comment regarding line 47-50.
Summary and expectations
I am not sure what the purpose is of this section. You do not need to outline research questions but if you do, these need to be in the introduction. Please remove this section.
Methods
Line 164: United States of America (again, think readers not familiar with the setting)
Setting: This section should describe the setting (not how many interviews were conducted)
Data collection – please provide a section on data collection that outlines how participants were recruited, inclusion / exclusion criteria, consent, generation of the interview guide (how did you decide on the questions), example topics covered, duration of interviews, how many conducted. Please use appropriate methodological citations in this section.
Data analysis – please provide a section on data analysis, that outlines transcription process, a more detailed account of the analysis and interpretation process. Make it clear how you arrived at the themes that are presented in the findings. Please use appropriate methodological citations in this section.
Discussion
Line 575-586: This is not needed. You have said this. Just refer to what your study was about, your findings and get into the discussion.
Line 626-635: Try to embed some citations in here. You are making big important arguments, and given that this is a discussion, some citations should be used here to add further sophistication.
Author Response
We thank the editor and anonymous reviewers for the opportunity to revise and resubmit our manuscript to IJERPH. The reviewers provided us with thoughtful and thorough evaluations of our paper along with helpful comments. We believe that the revisions we have made based on this input have allowed us to develop an improved manuscript. While we feel that we have attended to each of the concerns raised during the review process, please note that if we have missed something, we are more than willing to make the necessary changes going forward. As requested, we have provided responses to the specific points raised by each reviewer.
Reviewer 2:
- Reviewer’s comment: Thank you for inviting me to review this manuscript. The style of writing is clear, sophisticated and consistent throughout. I believe the authors engaged in a very interesting topic, tackling a current issue that requires our continued attention to better support female collegiate athletes. I believe there is scope to further improve the manuscript and outline detailed comments below to guide a manuscript revision:
- Authors’ Response: We sincerely thank the reviewer for their generous and encouraging feedback. We appreciate the recognition of the manuscript’s clarity and scholarly contribution to understanding the experiences of female collegiate athletes. In response to the reviewer’s detailed and constructive comments, we have carefully revised the manuscript to improve its organization, strengthen methodological transparency, and clarify key arguments throughout.
- Reviewer’s comment: Line 40-41: The sentence is not easy to follow. Do you mean that 56% of collegiate athletes are male?
- Authors’ Response: Thank you for noting this ambiguity. We have revised the sentence for clarity to read: “While approximately 56% of NCAA student-athletes are male, nearly a quarter of a million women (233,662) participated in college athletics during the 2023–2024 academic year [1].” This revision clarifies the statistic and improves readability.
- Reviewer’s comment: Line 41-42: negative impact of all sport? Or just collegiate sport? This needs to be made clearer.
- Authors’ Response: We appreciate this observation and have clarified that our focus is specifically on collegiate athletics. The revised sentence now reads: “However, the literature examining how participation in collegiate sports might negatively affect athletes disproportionately emphasizes male athletes, marginalizing the experiences of their female counterparts.” This edit specifies the level of sport under discussion and improves precision.
- Reviewer’s comment: Line 47-50: The case for what the challenges are associated with collegiate sport need to be made clearer. At the moment you say that there is a lot of pressure but without explanation of what the problems are. What has the literature reported here? If non-USA readers pick this up (I am one), they will not know why there are problems with collegiate sport. Explaining this is important to set the scene and allow you to explain why (and how!) these issues are amplified for female athletes.
- Authors’ Response: Thank you for this valuable feedback. We have expanded this section of the Introduction to clarify the challenges associated with participation in U.S. collegiate athletics, drawing from the cited literature on student-athlete stress, academic pressures, and mental health concerns. The revised passage now highlights that collegiate athletes often face competing academic and athletic demands, physical strain, injury, and pressure to maintain scholarships and eligibility—all of which can contribute to psychological distress. We then emphasize that these stressors may be amplified for female athletes due to gendered power dynamics, resource disparities, and coaching relationships. These revisions provide clearer context for both U.S. and international readers.
- Reviewer’s comment: Line 73-75: remove, not needed. You could say that you used qualitative methodology to highlight / examine x y z.
- Authors’ Response: Thank you for this suggestion. We have removed the referenced lines and streamlined the text.
- Reviewer’s comment: Line 76-85: I am not sure what this last paragraph is trying to achieve. Some of this needs to be earlier on where you want to write about the challenges of collegiate sport. Others, about the CAR, is somewhat repetitive of what you already said. After your aim and objectives, all you want to say why this study is significant in a short 2-3 lines. The gap (which I feel this paragraph is trying to emphasise) should be clear before this. So please revise. Especially seeing that you then have whole section on the CAR after the introduction.
- Reply: Thank you for this valuable feedback. We have revised the Introduction to integrate the discussion of challenges in collegiate sport and the significance of the coach–athlete relationship earlier in the section, removing repetition with the later “Coach–Athlete Relationships” discussion. The Introduction now concludes with a concise statement of the study’s significance and aims. These revisions clarify the study’s focus and improve overall flow, as the reviewer suggested.
- Reviewer’s comment: Coach-Athlete Relationships: Helpful, or Harmful? I like this section but I think this could be placed in the introduction, without sub-heading, just in the narrative and replace some of the things you currently say. I think this section also answers the questions I had in the introduction – see my comment regarding line 47-50.
- Authors’ Response: We appreciate the reviewer’s positive feedback on this section. In response to another Reviewer’s prior guidance, we have retained “Coach–Athlete Relationships: Helpful, or Harmful?” as a subsection within the Introduction rather than removing the subheading. This structure ensures clarity and consistency with the journal’s format while maintaining the flow between the background literature and the study’s conceptual framing. We have, however, slightly refined the transition to improve narrative continuity between the main Introduction and this subsection.
- Reviewer’s comment: Summary and expectations: I am not sure what the purpose is of this section. You do not need to outline research questions but if you do, these need to be in the introduction. Please remove this section.
- Authors’ Response: Thank you for this helpful suggestion. We have removed the Summary and Expectations section and incorporated the relevant content into the Introduction, where the study’s aims and central research question are now clearly stated.
- Reviewer’s comment: Methods: Line 164: United States of America (again, think readers not familiar with the setting)
- Authors’ Response: Thank you for this helpful reminder to consider an international audience. We have revised the sentence for clarity to read: “This research involved interviews with former female NCAA student-athletes who competed and currently reside in the United States of America (USA).” This change ensures the context is clear to readers unfamiliar with the U.S. collegiate athletic system.
- Reviewer’s comment: Setting: This section should describe the setting (not how many interviews were conducted)
- Authors’ Response: Thank you for this helpful clarification. We have revised the Setting section so that it now focuses on the broader research context—the NCAA structure, participant population, and rationale for selecting former U.S.-based athletes—rather than on procedural details. Information about the number and logistics of interviews has been moved to the Measures and Procedures subsection, ensuring that the Setting section now accurately describes the study context rather than data collection specifics.
- Reviewer’s comment: Data collection – please provide a section on data collection that outlines how participants were recruited, inclusion / exclusion criteria, consent, generation of the interview guide (how did you decide on the questions), example topics covered, duration of interviews, how many conducted. Please use appropriate methodological citations in this section.
- Authors’ Response: Thank you for this detailed feedback. We have restructured the Methods section to include distinct subsections—Setting, Recruitment, Data Collection, and Data Analysis—to clarify the research process. The revised Data Collection subsection now outlines participant recruitment, inclusion criteria, consent procedures, the development of the interview guide (based on prior literature), key interview topics, number and duration of interviews, and relevant methodological citations. These revisions provide clearer organization and greater transparency in accordance with the reviewer’s recommendation.
- Reviewer’s comment: Data analysis – please provide a section on data analysis, that outlines transcription process, a more detailed account of the analysis and interpretation process. Make it clear how you arrived at the themes that are presented in the findings. Please use appropriate methodological citations in this section.
- Authors’ Response: Thank you for this helpful suggestion. We have expanded and clarified the Data Analysis subsection of the Methods to describe the transcription process, coding procedures, and thematic development in greater detail. The revised section now specifies how transcripts were generated, reviewed, and de-identified; how open codes were created and refined through iterative, consensus-based analysis; and how thematic saturation was determined. We have also included appropriate methodological citations (e.g., Charmaz, Brinkmann & Kvale) to support the qualitative approach and analytic rigor. These revisions make our analytic process and interpretation of themes more transparent and aligned with best practices in qualitative research.
- Reviewer’s comment: Line 575-586: This is not needed. You have said this. Just refer to what your study was about, your findings and get into the discussion.
- Authors’ Response: Thank you for this observation. We have removed the redundant introductory lines from the beginning of the Discussion section and streamlined the text to transition directly from a brief statement of the study focus to the interpretation of key findings. The revised Discussion now begins by situating the results in relation to existing literature rather than restating the study background.
- Reviewer’s comment: Line 626-635: Try to embed some citations in here. You are making big important arguments, and given that this is a discussion, some citations should be used here to add further sophistication.
- Authors’ Response: Thank you for the suggestion. We have added supporting citations to this section of the Discussion to strengthen and contextualize the arguments presented.
Round 2
Reviewer 1 Report
Comments and Suggestions for Authors Dear Editor, Thank you for the opportunity to review the paper. The authors responded to my questions and revised the manuscript accordingly. I recommend the paper titled "Coach-Athlete Relationships and Mental Health: An Exploratory Study on Former Female NCAA Student-Athletes" for acceptance in the prestigious IJERPH.Good luck.